# Cancer Immunotherapy Dosing: A Pharmacokinetic/Pharmacodynamic Perspective

**DOI:** 10.3390/vaccines8040632

**Published:** 2020-10-31

**Authors:** Félicien Le Louedec, Fanny Leenhardt, Clémence Marin, Étienne Chatelut, Alexandre Evrard, Joseph Ciccolini

**Affiliations:** 1Institut Claudius-Regaud, Institut Universitaire du Cancer de Toulouse (IUCT)-Oncopole, and Cancer Research Center of Toulouse (CRCT), Inserm U1037, University of Toulouse, 31100 Toulouse, France; lelouedec.felicien@iuct-oncopole.fr; 2Institut de Cancérologie de Montpellier (ICM) and Institut de Recherche en Cancérologie de Montpellier (IRCM), Inserm U1194, University of Montpellier, 34090 Montpellier, France; Fanny.Leenhardt@icm.unicancer.fr; 3Assistance Publique—Hôpitaux de Marseille (AP-HM) and Simulation Modeling Adaptive Response for Therapeutics in cancer (SMARTc), Centre de Recherche en Cancérologie de Marseille (CRCM), Inserm U1068, Aix Marseille University, 13009 Marseille, France; clemence.marin93@gmail.com (C.M.); ciccolini.joseph@gmail.com (J.C.); 4Centre Hospitalier Universitaire de Nîmes Carémeau, Nîmes, France and IRCM U1194, University of Montpellier, 34090 Montpellier, France; alexandre.EVRARD@chu-nimes.fr

**Keywords:** checkpoint inhibitors, pharmacokinetics, dosing

## Abstract

Immune check-point inhibitors are drugs that are markedly different from other anticancer drugs because of their indirect mechanisms of antitumoral action and their apparently random effect in terms of efficacy and toxicity. This marked pharmacodynamics variability in patients calls for reconsidering to what extent approved dosing used in clinical practice are optimal or whether they should require efforts for customization in outlier patients. To better understand whether or not dosing could be an actionable item in oncology, in this review, preclinical and clinical development of immune checkpoint inhibitors are described, particularly from the angle of dose finding studies. Other issues in connection with dosing issues are developed, such as the flat dosing alternative, the putative role therapeutic drug monitoring could play, the rise of combinatorial strategies, and pharmaco-economic aspects.

## 1. Introduction

Immune check-point inhibitors (ICIs), considered as a breakthrough innovation in oncology, are characterized by the fact that their impact in terms of long-term survival remains restricted to a small subset (20–40%) of patients in an even smaller number of cancer diseases such as melanoma, lung, head-and-neck, and kidney cancers. For example, in some cases, sustained biological response can be reported after a single injection, i.e., long after the drug has disappeared from the body; in other cases, complete remission is achieved in advanced disease with once dismal prognosis, whereas elsewhere, no therapeutic effect or toxicities is observed. Importantly, robust and valid biomarkers with ICIs are yet to be found. For instance, PD-L1 expression, tumor mutational burden, neutrophils-to-lymphocytes ratio, history of corticoids or antibiotics use, plus rising evidences on the importance of gut microbiota have all been associated with ICI efficacy. However, none of them has emerged as a fully consensual predictive tool thus far [1]. This lack of available and validated biomarkers for forecasting clinical outcomes is a major concern, especially with respect to the high cost of ICIs. Consequently, the marked pharmacodynamics (PD) variability observed at bedside calls for reconsidering to what extent approved dosing in clinical practice is optimal or whether efforts for customization should be made for some patients. Indeed, there is a rising interest for dose individualization approaches in oncology, especially with the therapeutic drug monitoring of several cytotoxics [2] and oral targeted therapies [3]. To better understand whether dosing could be an actionable item in oncology, here, we review the preclinical and clinical development of ICIs. In addition to dose-finding studies, other related issues such as the trend towards flat dosing, therapeutic drug monitoring (TDM), combinatorial strategies, and pharmaco-economic aspects are developed as well.

## 2. Preclinical Models in the Era of Immunotherapy: Challenges and Pitfalls

Nonclinical studies are designed to check the safety and efficacy of drug candidates prior to early clinical trial phases—as well as to provide clues for the starting dose in first-in-human (FIH) studies. Regulatory safety studies (i.e., safety pharmacology, toxicokinetics, two-week/four-week toxicology, chronic toxicology, reproductive toxicology, search for genotoxicity, and carcinogenicity) must comply with strict guidelines in order to obtain Investigational New Drug (IND) status [4]. These studies are first performed in vitro, then in vivo in small species (i.e., rodents), and then in large species (i.e., dogs or non-human primates). On the contrary, efficacy studies can be performed using a wide range of in vitro and in vivo models depending on the therapeutic class of the compounds being tested, with few guidelines other than to use “the most appropriate model.” In oncology, countless in vitro models exist using cancer cell lines cultivated according to 2D or 3D (spheroids) protocols, along with more sophisticated models using cancer cells enriched with fibroblasts or endothelial cells [5], finally leading to complex human organoids [6]. Similarly, in vivo, almost all kind of tumors can be tested, mostly in small rodents, after ectotopic (i.e., subcutaneous) or orthotopic (i.e., in the organ of origin) engraftment. In oncology, in vivo models are mostly based on human tumor xenografts from established cell lines or from patient biopsies (patient-derived xenografts—PDXs) in order to better mimic human pharmacology when testing later active compounds. To this end, several strains of immune-compromised mice have been successfully developed over the last few decades, ranging from animals with partly abolished immune systems (e.g., NMRI mice, Swiss nude mice) to totally immuno-compromised animals (e.g., Nod-Scid Gamma mice). 

Since ICIs do not exert direct anti-proliferative activity on cancer cells, but are instead expected to harness tumor immunity, either by targeting cytotoxic T-lymphocyte antigen 4 (CTLA-4) or the programmed cell death protein 1/programmed death ligand 1 (PD-1/PD-L1) axis, human xenografts in immuno-compromised mice are no longer a suitable model. This has led investigators to shift toward immune-competent syngeneic mouse models with spontaneous or implanted murine tumors, thus making it possible to study the impact of the tested compounds on tumor immunity and to search for subsequent anti-proliferative effects. Non-clinical experiments with drug candidates in immunotherapy focus mostly on deciphering the pharmacology of the targeted pathways, assessing cytokine release potential, and studying receptor occupancy by using the models most likely to mimic tumor immunity in humans. Unfortunately, murine tumors (either grafted or spontaneously obtained in specific mutated strains) are not suitable to then test the actual developed drugs—first, because of slight differences in murine CTLA-4 or PD-1 targets and their human counterparts; and second, because all ICIs are human or humanized monoclonal antibodies (mAbs), against which the murine immune system is likely to react by producing neutralizing anti-drug antibodies. This means that drug candidates can no longer be directly administered, leaving only their murine avatars, created specifically against their respective murine targets. Unfortunately, this likely induces further experimental biases due to the marked differences between murine mAbs and human mAbs in terms of pharmacokinetics (PK) and target engagement. 

To resolve this inherent difficulty, sophisticated animal models such as human knock-in (KI) mice, immuno-avatar mice, hemato-lymphoid humanized mice or immune-PDX mice have been developed [7], making it possible to test actual drug-candidates in mouse models combining human xenografts with relevant, humanized immunity, and stroma cells. Such models at least partially respond to some of the issues involved when using murine clones of ICIs to be tested in animals. For example, human KI mice (i.e., animals bearing transgenes for human checkpoint molecules) make it possible to use actual human drugs, to study to what extent they further elicit cell-mediated cytotoxicity such as antigen-directed cell cytotoxicity (ADCC), and are also good models to mimic immune-related adverse events (irAEs) [8]. Immuno-avatar models are immuno-compromised mice engrafted with human peripheral blood mononuclear cells (PBMCs) that allow tumor xenografts to be later studied in a heterologous or autologous setting. However, because of the histocompatibility complex mismatch between murine cells and human T cells, xeno-reactions frequently occur with immuno-avatar models, leading some research teams to further deplete human CD4+ T-cells from PBMCs [9]. Hemato-lymphoid humanized mice are more complicated models in which immuno-compromised mice are transplanted with CD34+ human hematopoietic stem and progenitor cells and then receive human cytokines to generate proper immune cell functionalization [10]. However, as for immuno-avatars, human xenograft vs. host disease can show more frequent anemia since murine red blood cells are likely to be phagocytosed by human macrophages. The immune–PDX model is an alternative where the human stroma and relevant tumor–tumor-infiltrating-lymphocytes (TILs) interface is maintained, thus perfectly mimicking the human tumor microenvironment. However, the drawback of this model is that experiments must be carried out rapidly before innate murine cells start replacing human counterparts [11]. Consequently, because of the complex interplay between immune cells, drugs, tumor cells, and the tumor microenvironment with ICIs, and given the respective drawbacks of the above-mentioned models, predicting human efficacy or tolerance from animal observations remains particularly challenging in the era of immunotherapy [12]. Particularly problematic is the transposition of PK/PD relationships from animal studies to dose-finding studies in humans. Regarding FIH trials, there are no universal guidelines for dose selection using the minimum anticipated biological effect level (MABEL) approach. Preliminary data collected from experimental studies are essential for selecting the starting dose. Among them, studies on human and animal target cells, in vitro and in vivo concentration-effect studies, along with the most comprehensive knowledge on animal vs. human differences on dose/exposure relationships and drug disposition, target expression, affinity of target-binding, and intrinsic efficacy, duration and reversibility of effect. Starting dose selection is usually based on evidence for cytokine release (e.g., from isolated human PBMCs studies), in vivo toxicology for non-observed adverse effect level (NOAEL) determination (e.g., with 50x safety factor using standard cyno-tox models, as for atezolizumab), and receptor occupancy data [13]. Once ICIs have been approved, the shift from mg/kg dosing to flat-dosing can be made, primarily on the basis of human PK/PD studies [14] with extensive pharmacometric support, such as with pembrolizumab (see next section) [15].

## 3. Early Clinical Trials and Dose-Finding Studies

Early clinical trials consist primarily in evaluating the safety of a drug candidate, focusing on its fate in the human body (i.e., PK), studying possible adverse effects in humans (i.e., safety), and obtaining first evidence of PD effects (i.e., target engagement as a surrogate of therapeutic efficacy). Usually these data are collected from healthy volunteers, but in oncology for ethical reasons these early studies are mostly carried out in cancer patients, given possible safety concerns [16]. Ultimately, early clinical trials must define the range of doses that will be tested in further phase I trials (e.g., single-ascending dose and multiple-ascending dose phases I), thus eventually leading to the recommended phase-2 dose (RP2D) for larger, phase-2 efficacy studies. RP2D must be confirmed during registration of comparative phase-3 studies, prior to final approval.

Of note, historically with anticancer agents, optimal RP2D has long been defined as the level triggering severe but reversible toxicities in no more than 30% of the patients, a level best known as the maximum tolerated dose (MTD). It must be underlined that this definition barely applies to the most recent oral targeted therapies and biologics, including ICIs, since MTD is at times never even reached in standard dose-ranging studies. For example, escalating studies with a single agent (nivolumab, ipilimumab, or pembrolizumab) have thus far not made it possible to identify MTD for these drugs [17]. In addition, the search for MTD is historically based upon the “the more you administer, the more you get” assumption driven by the Hill equation. With respect to the very mechanism of action of ICIs, expected to harness tumor immunity without exerting direct cytotoxic activity, and because immune-related adverse effects are more influenced by patients’ immunity and individual susceptibility than by actual drug dosing and exposure levels, defining MTD is no longer relevant in the era of immunotherapy. As a consequence, early trials with ICIs do not seek to determine MTD, but rather RP2D, primarily based on target engagement considerations.

As for all other biologics, ICIs must comply with the MABEL approach when entering FIH studies. It should be noted, however that this approach does not make it possible to determine a starting dose. The latter is based on all data collected in non-clinical studies, including—but not limited to—in vitro pharmacology data from target cells, concentration-effect studies, and all animal–human comparison data available for drug PK; target affinity, duration, and reversibility of effects; and PK/PD relationships [16]. For example, the NOAEL of anti-PD-1 nivolumab and anti-CTLA-4 ipilimumab were, respectively, 50 mg/kg and 10 mg/kg in cynomolgus monkeys, respectively, while MABEL-based starting doses during FIH studies were, respectively, 0.1 mg/kg for nivolumab and 0.3 mg/kg for ipilimumab in humans [18,19]. The discrepancy in the correction factors used between NOAEL dosing and actual starting-doses in humans (i.e., a 500 factor for nivolumab and only a 30 factor for ipilimumab) encapsulates all the differences between these two drugs in terms of animal-to-human transposition for target engagement, PK/PD relationships, and immune-related toxicities. It also highlights how starting doses cannot, a priori, be uniformly defined. 

For example, the first dose-finding studies for blockbuster anti-PD-1 pembrolizumab included cohorts of 1 mg/kg, 2 mg/kg, and 10 mg/kg every two weeks (Q2W). As with nivolumab and ipilimumab, no dose-limiting toxicities were encountered at these dose levels, and no MTD was identified. Based on PK assessment and the 26-day half-life measured in humans, the dosing interval was extended to Q3W. Next, further intra-patient dose escalation was performed for dose ranging, where patients were escalated in three steps (i.e., day 1, 8, and 22) from very low doses (i.e., 0.05–0.06 mg/kg) to high doses (i.e., 2 mg/kg and 10 mg/kg) [20]. To assess target engagement, interleukin 2 (IL-2) ex-vivo assay was used as a surrogate to determine the effective pembrolizumab concentration likely to inhibit PD-1. Staphylococcal enterotoxin B induces lymphocyte IL-2 release and the active PD-1 pathway that normally inhibits this release. Blocking PD-1 lifts the inhibition, so measuring IL-2 release thus provides data on the PD-1 blockade: the greater the IL-2 release, the stronger the PD-1 inhibition. For pembrolizumab, 10 µg/mL were enough to achieve 95% inhibition of PD-1, and translational PK/PD modeling showed that 1 mg/kg Q3W led to a 64% probability of PD-1 blockade, establishing this level as the lower boundary for clinical efficacy. Further PK/PD modeling helped to identify 2 mg/kg as the RP2D, i.e., safe dosing ensuring trough levels above the efficacy level with >90% probability of maximal target engagement [13,14]. Once the drug was finally approved at 2 mg/kg Q3W, further modeling was used to propose a 200 mg Q3W flat-dosing [21].

With nivolumab, a seminal early phase study testing doses ranging from 0.1 mg/kg to 10 mg/kg showed that 60–70% target engagement (evaluated PD-1 blockade measured in circulating T lymphocytes) was achieved, regardless of the dosing [22,23]. Further phase I studies led to an RP2D for nivolumab at 3 mg/kg Q2W, and as with pembrolizumab, nivolumab dosing was next shifted to 240 mg Q2W or 480 mg Q4W flat-dosing, mostly based upon in silico PK/PD modeling [24]. 

Dose-finding studies based on a combinatorial regimen with ICIs are tricky to set up, given the lack of identified MTD for single agents. Several designs such as the canonical 3+3 up and down or 6+6 up and down dose escalation studies can be undertaken. In addition, more sophisticated approaches such as those using the modified toxicity probability intervals charts can be used [25]. When combining several ICIs, such as the ipilimumab + nivolumab combo in melanoma and non-small cell lung cancer (NSCLC), a limited range of dosing has been tested (i.e., 3mg/kg and 1 mg/kg and vice-versa). Since both drugs were approved as single agents at 3 mg/kg, that no MTD was identified in their respective dose-ranging studies, and that ipilimumab was the core component of the combination for melanoma and nivolumab for lung cancer, 3 mg/kg was used as a reference. Previous phase-1 studies had already suggested that 1 mg/kg could lead to adequate target engagement [26,27]. Of note, most combinatorial regimens with ICIs (e.g., with radiation therapy, cytotoxics, or targeted therapies) are primarily based on empirical and underpowered strategies to determine respective component dosing, possibly explaining the high attrition rate of current combination studies with immunotherapies [28]. 

Additionally, checkpoint inhibitors that target other axes than CTLA-4 or PD-1/PD-L1 are currently being developed [29]. Of note, none of them have reached the shelves yet, and late clinical phases are yet to come. In FIH studies of Hu5F9-G4, an inhibitor of phagocytosis checkpoint CD47, dose selection was based on receptor occupancy of CD47 at the surface of red blood cells or white blood cells [30,31]. Here, full receptor occupancy could only be observed at high doses, i.e. 10 to 30 mg/kg weekly. Generally speaking, mAbs get internalized and catabolized after the fixation to their target. This can be a non-linear elimination pathway for mAbs and is best known as target-mediated drug disposition (TMDD) [32]. Due to the ubiquity of CD47, this process was particularly important for Hu5F9-G4, requiring important doses to saturate this so-called “antigen sink” and observe a linear PK [33]. A particular attention to the potential clinical relevance of this “antigen sink” in terms of PK/PD relationships should be paid during the future clinical development of this new generation of ICIs.

## 4. Flat Dose and Modified Schedules

Most ICIs (except for atezolizumab and cemiplimab) were initially developed and approved by the Food and Drug Administration (FDA) as body weight (BW)-based dosing regimens (Table 1). This choice was primarily dictated by usual practice with anticancer drugs for which metrics associated with patient’s corpulence such as body surface area (BSA)-dosing or BW is used. BSA and BW are closely correlated, and the main rationale for BSA- or BW-based dosing assumes that drug clearance (CL) is proportional to BW [34]. Although the correlations between morphological characteristics and drug CL are only weak, BSA-dosing practice remains a standard way to administrate cytotoxics [35,36]. However, phase 1 studies with ICIs showed a wider therapeutic index. Side effects were poorly related to the dose (cf 5. adverse events and dose section), which encouraged considering flat dosing ensuring maximal target engagement. Flat dose presents several advantages such as facilitating prescription, improving capacity planning of production of hospital pharmacies, and reducing drug wastage and patient waiting time [37]. In order to justify this switch from individual BW-dosing to a unique flat dose, population PK analyses of data accumulated during drug development were performed for avelumab [38], durvalumab [39], nivolumab [40] and pembrolizumab [41]. First, they revealed that BW contributes only marginally to inter-individual variability (IIV) of PK parameters, particularly the CL. Secondly, they made it possible to simulate flat dosing and then to compare simulated plasma ICI concentrations to actual concentrations corresponding to a mg/kg dose [21,39,42,43]. Flat dosing was not associated with a larger IIV in plasma concentrations, and led to statistically similar exposures, regardless of the dosing. For instance, in a population PK analysis of 7407 plasma durvalumab concentrations obtained from 1409 patients included in phase 1/2 and phase 2 studies, BW was identified among covariates correlated with durvalumab CL (i.e., durvalumab CL was proportional to BW^0.389^) [39]. However, simulations of flat-dosing regimens, 750 mg Q2W and 1500 mg Q4W vs. 10 mg/kg Q2W indicated that both regimens yield similar median steady-state exposures and variability, with no increased incidence of extreme concentration values with flat-dosing as compared with weight-based dosing regimen. 

Modeling and simulation has also allowed considering changes in administration schedule with larger time-intervals between administrations. Long et al. [44] performed analysis of anti-PD1 nivolumab pooled data from 3817 patients treated according to several schedules: 3 mg/kg Q2W, 240 mg Q2W, 480 mg Q4W, or 3 mg/kg Q2W followed by 480 mg Q4W schedules. First, nivolumab 480 mg Q4W resulted in an only modest increase in IIV in exposure relative to 3 mg/kg Q2W dosing (coefficient of variation [CV], 46.4% and 43.4%, respectively). In addition, the variability in plasma nivolumab concentrations was limited, as shown by comparison of mean (CV%) minimum and maximum steady-state concentrations with Cmin,ss of 65.7 µg/mL (52%) vs. 55.2 µg/mL (63%) and Cmax,ss of 127 µg/mL (45%) vs. 184 µg/mL (58%) for the 480 mg Q4W and 3 mg/kg Q2W schedule, respectively. The safety of the 480 mg Q4W regimen was then assessed in a clinical trial. Indeed, simulations found no differences between regimens on a population scale. However, for a given individual with extreme BW (e.g. 40 kg), flat-dosing would represent a substantial increase (+100%) in exposure to nivolumab compared to BW-dosing. The incidence of treatment-related adverse events was similar between the different schedules and between BW groups, indicating that the higher exposure of patients in the lower BW group treated by flat dose was not associated with higher risk of side effects. Moreover, the 61 patients who shifted from nivolumab 3 mg/kg Q2W to 480 mg Q4W showed no increased incidence of adverse events after the switch. This highlights how tolerance to most ICIs is not related to exposure levels but rather to patient’s innate immunity and susceptibility. Excepted anti CTLA-4 ipilimumab [45], no exposure/toxicity relationships have been evidenced with ICIs, especially the ones interacting with the PD-1/PD-L1 axis (see the next paragraph). 

For anti-PD-1 pembrolizumab, similar simulations have been performed based on PK data and dose-response relationships observed over five separate randomized dose comparisons (2 mg/kg vs. 10 mg/kg) and three separate randomized schedule comparisons (Q2W vs. Q3W) [21]. Flat dose-response and exposure-response relationships, with similar efficacy and safety outcomes over this five-fold dose/exposure range, first made it possible to replace 2 mg/kg of pembrolizumab Q3W by 200 mg Q3W and second to conclude that the pembrolizumab 400 mg Q6W dosing regimen is expected to be as effective and safe as the previous schedules. Simulations have shown that only a very small fraction of patients (≈0.5%) are expected to have a decreased Cmin after 400 mg Q6W, as compared with the Cmin observed with 2 mg/kg Q3W dosing [15]. Of note, atezolizumab and cemiplimab were immediately approved with fixed 1200 mg Q3W [46] and 350 mg Q3W [47] dose regimens, respectively.

## 5. Adverse Events and Dose

The safety of ICIs differs significantly from systemic cytotoxic chemotherapy, oral targeted therapies, or other mAbs targeting a specific receptor on tumor cells. Pharmacologists are puzzled when studying the dose/exposure/toxicity relationships with ICIs because neither the traditional concept of dose-intensity and MTD nor the more recent pharmacology of targeted therapies (mAbs or oral kinase inhibitors) applies [48]. The so-called immune-related adverse events (irAEs) are now well described clinically, but the predominant mechanism of action and the long-term effects on patients’ immunity remain unclear [49]. Adverse events of both CTLA-4 and PD-1/PD-L1 inhibitors come from non-specific activation of the immune system. This can occur through T-cell and B-cell activation and/or proliferation, increased cytokine release, cross-reactivity of tumoral antigens with normal tissues or direct effects of the mAbs [50]. For example, skin toxicity such as vitiligo occurs predominantly in melanoma patients, illustrating cross-reactivity of T-cells against tumor cells and normal tissues. Endocrine side effects are linked to CTLA-4 expression in the pituitary gland leading to hypophysitis, mainly in patients treated by ipilimumab and not by PD-1 or PD-L1 inhibitors [51]. Some adverse events are very frequent such as gastrointestinal (e.g., diarrhea, colitis), dermatological (e.g., pruritus, rash and vitiligo), or endocrine (e.g., hypophysitis, thyroid dysfunction) toxicities. Others can be less frequent, such as cardiac toxicity (<1%), but are sometimes fulminant and potentially life-threatening [52]. Because of the particular pharmacological mechanisms of upregulation in various immune pathways, irAEs could virtually occur in any organ of the body, and a comprehensive description of all possible irAEs can be found elsewhere [48,49,50]. Some clinical data describe a tumor-specific pattern of irAEs, suggesting an organ-specific microenvironment that could explain observed differences between patients treated by the same ICI but for different tumor localization [53]. Overall, irAEs occur frequently with a pooled incidence ranging from 54% to 76% according to a recent meta-analysis, regardless of toxicity grading [54]. Severe toxicities (> grade 3 of the Common Terminology Criteria for Adverse Events classification) occur in about 30% of patients for anti-CTLA-4 and about 15% for anti-PD-1/PD-L1, but prevalence of severe irAEs rises up to 50% in case of ICIs combination or in patients with a history of preexisting autoimmune disorders [55]. The onset of irAEs usually occur during the first four weeks after beginning treatment, however some long-term irAEs (i.e., >1 year) linked to the persistence of autoreactive T-cells can occur long after the completion of therapy [56]. The distribution of the type of irAE may vary according to gender and are more frequent in elderly patients [57,58].

The clinical development of previous mAbs in oncology has demonstrated that dose selection cannot be based on usual safety standards, since biologics are usually well tolerated. However, the “London catastrophe” with the CD28 superagonist TGN1412 highlighted the fact that pharmaco-modulation of the immune system could not be done empirically, leading to the use of adaptive models and the MABEL approach to select the appropriate starting dose during FIH studies [59]. Not surprisingly for ICIs, as seen above, MTD cannot be achieved in either non-clinical or clinical studies, but the relationship between dose/exposure and adverse events has been partially addressed in the clinical development phase and post-market use [45]. The very first observations with ipilimumab demonstrated a significant relationship between exposure (Cmin,ss) and safety with a significant increase of irAEs in higher Cmin,ss for doses ranging from 0.3 mg/kg to 10 mg/kg [60]. A similar trend was observed with the other anti-CTLA-4 inhibitor tremelimumab, although very few PK studies are available in a real-world setting since the drug has failed to meet primary endpoints in most clinical trials [61]. In contrast, a flat relationship between dose/exposure and adverse events was observed for anti-PD-1/PD-L1 immunotherapies regardless of dosing and scheduling. Indeed, no relationship was found between exposure (average concentration at steady-state or AUC) and grade > 3 irAEs for ICIs interacting with the PD-1/PD-L1 axis such as nivolumab, pembrolizumab and durvalumab [45]. 

The next challenge is to clarify whether some irAEs could be predictive of outcome and used as a surrogate marker of efficacy in a toxicity-driven strategy, as previously described for anti- epidermal growth factor receptor (EGFR) mAbs. There is growing evidence that the “no pain, no gain” paradigm is still true with ICIs, and that patients displaying irAEs are more likely to benefit from ICIs, suggesting that the pathophysiology of anti-tumor effects and some adverse events may share the same mechanism of action [62,63,64]. The most consistent data supporting this hypothesis were obtained with anti-PD-1/PD-L1 inhibitors and dermatological adverse events such as vitiligo and rash occurring more frequently in patients with better progression-free survival (PFS), overall survival (OS) and overall response rate [65]. Of note, a possible confounding factor when deciphering the links between irAEs and antitumor efficacy is the fact that the longer the survival, the longer the treatment, and therefore the higher the risk of experiencing treatment-induced toxicities. In addition, not all irAEs are a necessary evil, first because they are not always associated with optimal response and, second, because they can lead to irreversible organ damage, especially of the endocrine system, strongly hampering quality of life and delaying ICI (re)administration. Combination therapy of CTLA-4 and PD-1/PD-L1 inhibitors significantly enhances the occurrence of irAEs with an overall fatal outcome ranging from 1 to 2%, mainly due to myocarditis and colitis. Since only a segment of patients will benefit from immunotherapy, several potential biomarkers have been identified that can help avoid overtreatment and unnecessary toxicity [66]. For instance, a weak baseline of IL-6 serum level has been associated with higher rates of irAEs [67]. Serum levels of others soluble biomarkers, cytokines, or auto-antibodies have been associated with irAEs in several studies and may translate to clinical routine in the near future [68]. However, from the pharmacologist’s point of view, it is still not clear whether irAEs, at least in part, are a drug exposure matter or not. Indeed, except for the notable exception of ipilimumab, we still lack PK analyses to untangle the role of dose and drug exposure in irAEs occurrence from the role of PD side effects unrelated to drug concentration. 

## 6. Is There Some Room for TDM with ICIs?

As for other drugs, and particularly anticancer compounds, it is critical to determine the appropriate PK metrics associated with PD endpoints. For cytotoxics, both hematotoxicity and efficacy are correlated with the whole plasma exposure, i.e. area under the curve of plasma-concentrations vs. time (AUC). For small targeted therapy molecules that are given daily, trough levels (i.e., residual concentrations) at steady-state and also represents usually the best PK metrics for PD. For mAbs targeted therapy, as for most ICIs, phase 1 trials have revealed a rather flat dose-response. The only notorious exception is ipilimumab, an anti-CTLA-4, for which a clear dose-dependent PD has been observed [60]. Although a wide PK IIV has been observed for all of these mAbs with a ratio around ten between lowest and highest CL values, these flat-responses have not made it possible to identify a particular PK metric (i.e., Cmax, Cmin, AUC) correlated with PD endpoints. It should be noted that the comparison of several schedules (weight-based dosing vs. flat-dosing, Q6W vs. Q3W, …) described in the previous paragraph were based on the hypothesis that Cmin,ss and AUC are the potential PK drivers of efficacy (and Cmax,ss of safety). However, the data obtained during the development of the ICIs did not frankly confirm these hypotheses; with the exception of ipilimumab, no clear exposure–response relationships were observed when considering the range of plasma concentrations usually observed in patients. These observations do not advocate implementing TDM with immune ICIs, since even the lowest concentrations found in plasma are expected to be sufficient to ensure maximal target engagement.

Nevertheless, studies conducted in real-world settings have raised the question of possible exposure-response relationships with nivolumab in treating NSCLC. In a cohort of 76 patients, Basak et al. reported higher nivolumab trough concentrations in responders vs. non-responders [69]. No adjustment with disease-related covariates, such as performance status, were made in the latter analysis. However, these results were not confirmed by Bellesoeur et al. in a similar cohort of 81 lung cancer patients, since only performance status (Hazard ratio [HR] 1.85, 95% Confidence Interval [CI] 1.02–3.38) and baseline use of corticoids (HR 8.08, 95% CI 1.78–36.62) were found to be predictive of PFS [70].

The association of irAEs with PD-1/PD-L1 efficacy in the treatment of advanced urothelial carcinoma could have been considered a second argument in favor of the implication of higher plasma exposure in both efficacy and toxicity [71]. But this causal relationship between ICI concentrations and PD is doubtful. The most convincing study, Turner et al., compares the outcome of patients treated with 2 mg/kg or 10 mg/kg pembrolizumab for either advanced ipilimumab-refractory melanoma or previously treated PD-L1 positive NSCLC [72]. In each case, they observed a superposition of the curves describing survival probability vs. time for the first and fourth quartile CL at both doses [72]. Outcome was much better for those in the 1st quartile CL (i.e., patients with the lowest CL) than for those in the fourth quartile CL (i.e., patients with the highest CL), but was similar for each quartile subgroup regardless of the pembrolizumab dose (2 m/kg or 10 mg/kg). Should pembrolizumab exposure have had a direct impact on outcome, there would have been no superposition, but rather a better outcome at 10 mg/kg than at 2 mg/kg, provided that PK variability does not blur the picture. The authors considered that high CL is linked to the elevated catabolism induced by cachexia and is more a marker of refractory disease than a direct cause of treatment failure. This observation may be generalized to all ICIs [38,73,74,75], as well as for other mAbs [76,77,78]. Biological characteristics of cachexia, such as hypoalbuminemia, are associated with higher CL of ICIs [38,39,73,79,80]. These statistically significant correlations do not warrant individual dosing as a means to improve efficacy. First, clinical relevance is lacking (e.g., for durvalumab, low albumin levels gave a 22% reduction in AUCss for the 10th percentile level (30 g/L) compared to a typical patient [39]). Second, as stated before, the labeled doses are supposed to be associated with plasma concentrations (including Cmin) much higher than necessary to achieve maximal efficacy. Moreover, the time-decreasing CL generally observed with ICIs appears to be a consequence of the reversal of the cachexia syndrome [38,39,40,80]. This is consistent with the fact that this decrease in CL is greater in responding patients (as seen with nivolumab [75]) and is closely related to the normalization of albuminemia (as seen with durvalumab [73]). In some way, this change of PK parameters could be considered to be a marker of ICI efficacy. Some authors have developed machine-learning models based on the levels of several inflammatory cytokines (e.g., 8 and 16 for renal cell carcinoma [81] and advanced melanoma [82], respectively) to predict nivolumab CL, and eventually OS.

## 7. Are Combinatorial Regimens the Future of Immunotherapy?

Combining drugs is a key issue in cancer therapy. For cytotoxics, combining two or three drugs accounts for the vast majority of protocols. Small molecule targeted therapies, e.g. tyrosine kinase inhibitors (TKIs), are mainly used as single agents in treating monogenic cancer disease such as chronic myeloid leukemia, EGFR-mutated NSCLC, or advanced renal cancers. Phase 1 studies combining TKIs and cytotoxics have been performed in cancers not driven by a unique pathway, but these have often been associated with poor tolerance and sometimes with marked PK drug–drug interactions (DDI), thus limiting their use. Conversely, several combinations of a mAb with cytotoxics have resulted in a major increase in efficacy (e.g., anti-Her2 trastuzumab with taxanes or anthracyclins for Her-2 breast cancers [83]) or more modest improvement (e.g., anti-EGFR cetuximab [84] or anti-VEGFA bevacizumab [85] combined with cytotoxics for treating colorectal cancers). In immunotherapy, the objective of combinatorial strategies is to overcome the resistance to ICIs, whether it exists since treatment initiation (i.e. non-responding patients) or it appears after late relapse [86]. Several biomarkers have been proposed to predict efficacy to PD-1/PD-L1 inhibitors, the most important being tumor mutational burden and PD-L1 expression [87]. Thus, the ICI key questions for ICIs combination are: is it beneficial to combine immunotherapy drugs, and is there any possibility of “turning up the heat” of the tumor by using another therapeutic approach (i.e., radiation therapy, cytotoxics, or targeted drugs) to make it more sensitive to immunotherapy? Since ICIs are mAbs, their combination with other drugs (including other mAbs) is not likely to trigger PK DDI—an advantage compared to small molecules and cytotoxics that may interact with each other on liver metabolizing enzymes or drug transport) [88]. Indeed, PK of mAbs mostly relies on an unspecific intracellular lysosomal catabolism in the reticulo-endothelial system. Traditional processes such as glomerular filtration, membrane transporters or cytochrome P450-mediated metabolism are not implicated in mAbs disposition. It explains how scarce are DDI involving mAbs [89].

### 7.1. Combination of ICIs

The nivolumab/ipilimumab combination has been first approved for adult patients with advanced melanoma on the grounds of its improved efficacy. OS at five years was 52% for the nivolumab plus ipilimumab group, 44% for the nivolumab monotherapy group, and 26% for the ipilimumab monotherapy group) [90]. This combination was then approved for advanced renal cell carcinoma [91], for previously treated microsatellite instability-high (MSI-H)/mismatch repair deficient (dMMR) colorectal cancer [92], and, most recently, for hepatocellular carcinoma previously treated with sorafenib [93]. The greater efficacy, however, is associated with more frequent and more pronounced toxicity, requiring discontinuation of therapy in nearly 40% of patients. The combination of nivolumab and ipilimumab requires lower doses than when each drug is used in monotherapy. For example, 1 mg/kg nivolumab plus 3 mg/kg ipilimumab are used in metastatic melanoma, since too many dose-limiting toxicities were observed at 3 mg/kg nivolumab plus 3 mg/kg ipilimumab dosing [94]. Interestingly, MTD was not reached in single agent phase I trials (respectively up to 10 mg/kg and 20 mg/kg), thus illustrating the complexity of PD interactions between ICIs. 

### 7.2. Combination with Cytotoxics

Myelosuppression consecutive to administration of cytotoxics led to the widespread perception that cytotoxics are immunosuppressive and thus would not be ideal drugs for a combinatorial regimen with ICIs. However, cytotoxic treatment does not necessarily impact lymphocyte counts, contrary to neutrophil counts. Moreover, cell models have revealed several cytotoxic effects increasing ICI efficacy, such as direct stimulation of T-cells, maturation of dendritic cells (DC) and enhancing antigen presentation, and release of antigens [95]. The latter includes immunogenic cell death (ICD), a form of apoptosis that can induce an effective antitumor immune response through activation of DCs and the subsequent activation of specific T-cell responses. This has been shown for anthracyclines, topoisomerase I inhibitors and cyclophosphamide, to name a few [96]. Cisplatin has been shown to increase PD-L1 [97] expression and major histocompatibility complex class I expression on antigen-presenting cells [98]. Both docetaxel and paclitaxel have been shown to selectively decrease regulatory T-cells (Treg) and myeloid-derived suppressor cells in the tumor micro-environment (TME) and induce tumor infiltrating lymphocytes (TILs) [99]. In vivo, preclinical models have indicated that synergistic effects require a chemotherapy induction phase prior to ICI administration. Some combinations have already been approved, all as first-line treatment of metastatic NSCLC, including platinum compound, resulting in improved OS compared to chemotherapy alone [100,101]. All treatments are given concomitantly, irrespective of possible sequence-effect when combining drugs [27].

Numerous studies combining chemotherapy and ICIs are still underway. For most, all drugs are given at full dose the same day according to a Q3W schedule, reflecting that toxicity is not an issue because each drug has a specific toxicity profile. The ICI is administered as single agent maintenance phase until either progression occurs or toxicities become unacceptable. However, in some protocols, a chemotherapy induction phase is administered (e.g., with durvalumab after chemotherapy in NSCLC [102]) or, conversely, an immunotherapy induction phase. All of these clinical trials require biomarker analyses (e.g., PD-L1 expression before and after a chemotherapy induction phase) in order to better explain global clinical results and variability in tumor response. Finally, combining nanoparticles with immune checkpoint inhibitors is an appealing strategy, because nanoparticles could help to harness tumor immunity [103]. Recently, association between nab-paclitaxel and atezolizumab has proven to yield substantial increase in efficacy in triple negative breast cancer, either alone [104] or further combined with anthracyclines [105].

### 7.3. Combination with Targeted Therapy

Most combinations of ICIs with targeted therapy concern anti-angiogenic drugs, such as bevacizumab as a mAb against vascular endothelial growth factor (VEGF), or small molecule VEGF-receptor inhibitors such as axitinib and cabozantinib. This strategy consists in inducing vascular normalization by means of the anti-angiogenic drugs in order to restore immune cell functions [106]. Indeed, abnormal tumor vasculature in the TME is associated with impaired diffusion of the drugs (particularly mAbs and immune effector cells into the tumors) [107]. Moreover, it has been shown that angiogenic molecules such as VEGF prevent mobilization, proliferation, and the effector function of CD8-positive cytotoxic T lymphocytes, while promoting Tregs recruitment [107]. Several animal models have confirmed the synergistic activities of anti-PD-1/anti-angiogenic combinations, supporting several clinical trials, particularly for treating metastatic renal cancers where both antiangiogenic and ICIs have shown efficacy as a single agent. For example, axitinib associated with pembrolizumab [108] or avelumab [109] increased survival in advanced renal-cell carcinoma compared to sunitinib. However, some of these combinations, such as atezolizumab plus bevacizumab, are associated with a higher risk of treatment-related adverse effects apparently linked to antiangiogenic agents rather than to an increase of irAEs. Combinations of nivolumab or the nivolumab + ipilimumab combo with cabozantinib are also currently being investigated and showed that all drugs could be safely associated [110,111]. 

### 7.4. Combination with Radiotherapy 

The rationale for combining radiation therapy and ICIs is triple [112]. First, an abscopal response interaction is expected [113]. Radiation is expected to release tumoral antigens that would trigger an immunologic response, itself potentiated by ICIs. Thus, even the non-irradiated sites would benefit from the treatment. Second, immunotherapy is supposed to enhance the local in-field radiation response through both immune-adaptative and innate mediated CL of residual disease [114]. Surviving irradiated cells may represent targets of immune cells such as CD8 + cells. In vivo, upregulation, and increased expression of PD-L1 at the surface of tumor was also observed after fractionated radiotherapy [115]. Lastly, radiation therapy may help therapeutic mAbs to infiltrate tumor site through reduction in stroma density, a phenomenon best known as the radiation-enhanced permeation effect. The benefit of combining radiation and ICIs has been particularly highlighted for stereotactic body radiotherapy (SBRT) in treating early lung cancer or oligo-metastatic diseases, as well as stereotactic radiosurgery to treat brain metastases. The addition of high dose/fraction radiation increases the efficacy of the doublet therapy ipilimumab + nivolumab in treating melanoma patients with brain metastases. In a recent meta-analysis radiotherapy was not associated with additional grade 3–4 toxicities, independently of the radiation schedule (simultaneous or sequential) [116].

### 7.5. Other Combinations

Combinations of drugs from the three main groups of anticancer treatment have also been evaluated. For instance, atezolizumab plus chemotherapy plus bevacizumab has significantly improved PFS and OS of patients with metastatic non-squamous NSCLC, regardless of tumor’s mutational status and checkpoint expression [117,118]. ICIs have also been combined with other drugs. Ipilimumab with sargramostim (granulocyte-macrophage colony-stimulating factor or GM-CSF)—a cytokine increasing antigen presentation by dendritic cells and antitumor activity of T- and B-lymphocyte populations—have been associated with similar efficacy but, interestingly, with better tolerance of ipilimumab alone in metastatic melanoma [119,120]. Epacadostat failed to improve efficacy in combination with pembrolizumab in metastatic melanoma, despite promising early clinical results [121]. The dose of epacadostat might deserve consideration, since escalation was performed on this drug alone, with a fixed dose of pembrolizumab [122]. Finally, clinical trials combining tumor necrosis factor (TNFα) blocker and anti-PD-1 are currently being conducted (NCT03293784). Indeed, there is a strong rationale in the melanoma mouse model showing that TNF signaling impairs accumulation of CD8+ TILs [123].

## 8. The Pharmaco-Economic Aspects of Dosing Strategies

Biologics, and especially ICI mAbs, are very expensive drugs that could drastically hamper health care budgets at the hospital level. Avoiding spillage in preparation at hospital pharmacies could contribute to reducing treatment cost. Indeed, until 2017, nivolumab and pembrolizumab had weight-based dosage and dose adaptation for each cycle for every patient, representing both financial and human costs (e.g., pharmaceutical team, medical transports, hospital visits, etc.). Authors have tried to compare the cost between weight-based dosing, fixed dosing or, alternatively, dose-banding dosing. Hendriks et al. at the National Cancer Institute demonstrated in a real-world study that fixed dosing makes it possible to save about €3 million over a 15-month period [37]. In another study pre-fixed dosing reduced the risk of error, while saving time from prescription to production and reducing drug waste with shorter infusion and lower infectious risk [21]. Novel dosing strategies (fixed-dose vs. weight-based and frequency of administration) have been evaluated to conclude on efficacy, safety and pharmaco-economic balance. Single agent nivolumab used to treat melanoma and NSCLC, notably, was prescribed at 0.1 mg/kg or 0.3 mg/kg (melanoma) and 1 mg/kg or 3 mg/kg (NSCLC) once every two weeks. Cost and efficacy analyses concluded on the safe and effective use of 240 mg once every two weeks or 480 mg once every four weeks, regardless of patient weight [44]. The appropriate fixed dosage was estimated by major overlap of nivolumab exposure as a function of BW (BW between 34 kg and 180 kg). 

FDA-approved nivolumab fixed dosing based on both population PK and dose/exposure-response analyses demonstrating the same exposure, safety, and efficacy for the new dosing regimen (240 mg every two weeks). Early weight-based dose trials were performed for pembrolizumab (2 mg/kg and 10 mg/ kg, once every two or three weeks) in melanoma and lung cancer. Studies investigated the equivalence between doses at each weight that were found to be equally effective [124]. Based on these results, Merck performed subsequent phase 3 trials with a fixed dose of pembrolizumab (i.e., 200 mg) regardless of patient’s weight. These data suggest that fixed dosing has equivalent efficacy and safety [124]. But further analysis considering the average weight of cancer patients (75 kg), showed that only 150 mg of pembrolizumab would be actually required [125]. These FDA PK simulations demonstrated the same exposure between fixed and weight-based dosing, leading to changes in pembrolizumab dosing (200 mg once every three weeks or 400 mg once every six weeks). 

In response to the question of dose/efficacy evaluation, PK studies have been necessary to determine whether immunotherapy exposure-response relationships have been modified by dosing strategies. For pembrolizumab and nivolumab, PK modeling and simulation were performed, with cost analysis as endpoint. In Ogungbenro et al., relative cost was estimated for fixed and weight-based strategies, as well as PK-derived strategies for pembrolizumab. Each dosing strategy was evaluated using simulations (AUC, Cmin simulated, cost according to duration of treatment). The study concluded that the PK-derived dosing was the most cost-effective strategy [126]. However, a French study later concluded on the absence of a PK/PD relationship for nivolumab in NSCLC patients in the weight-based dosing cohort [70]. Further PK/PD studies are thus necessary to evaluate the role of TDM as a tool to improve the cost effectiveness of immunotherapy, if not for improving efficacy [40,127]. Nevertheless, PK variability must be taken into account to individualize immunotherapy dosing. In subgroup analyses based on body mass index (BMI), patients would appear to be more exposed at high BW (>130 kg) and less-exposed at low weight (<50 kg) in a weight-dose strategy; authors find opposite results with fixed dose for both nivolumab and pembrolizumab [126,128]. These findings would thus confirm fixed-dose overexposure for patients who are not overweight—a significant difference between American and European patients in clinical trials and real-world settings.

In a recent multicentric study of BMI in cancer patients treated by immunotherapy, authors concluded that dysregulated tumorigenic immune activity in the overweight could be reversed by immunotherapy. This observation can explain the observed longer OS in obese and overweight patients (BMI > 25) compared to BMI < 25, but the authors did not specify the dose strategy or the schedule of nivolumab and pembrolizumab [129]. The mean population weight is also to be taken into account in a pharmaco-economic analysis for immunotherapy dosing strategy, since mean European weight is lower than BW used in fixed-dose clinical trials, leading to extra cost per patient for European cancer centers in a fixed-dose strategy. In a pharmaco-economic evaluation of pembrolizumab dosing strategy in the United States, a 24% increase in expenses was highlighted for fixed-dosage compared to a weight-based strategy in NSCLC [130]. Mean pembrolizumab cost for weight-based strategy was evaluated at 3758 €/cycle (with 2 mg/kg for a 70-kg patient) and 5368 €/cycle for 200 mg for fixed-dose, with a three-weekly administration (2017 cost figures) [131]. In a French study, the extra mean cost for nivolumab between weight-based and fixed dose was evaluated. For the latter strategy, cost per infusion was estimated at €349, resulting in an increase in the national and annual budget for nivolumab of 16% (2017 data) [132]. These additional costs are not offset by the reduced frequency of administration or shorter infusion (for nivolumab at 480 mg) [133].

The financial impact of personalized dosing could potentially save approximately $2000 per dose of pembrolizumab. US health care could save approximatively $0.8 billion annually by using weight-based dosing for pembrolizumab when treating NSCLC patients [134].

Hospital pharmacy logistics must also be taken into consideration. Pembrolizumab preparation and administration has been estimated to be equal both in time and organization for fixed and weight-based dosing. Indeed, no remaining pembrolizumab is expected at 200-mg fixed dosing using 50- or 100-mg vials, while substantial amount of drug is discarded with weight-based dosing in pharmacies (respectively, 27 mg and 56 mg per patient, per administration would be wasted when using 50-mg and 100-mg vials) as shown by Freshwater et al. [21]. For pharmaco-economic studies, the cost of leftover drugs varies according to the market (i.e., available vial sizes). Bach et al. have proposed recourse to an additional vial size (10 mg) for nivolumab and pembrolizumab for weight-based dosing strategy to reduce wasting, with, respectively, an expected saving of $26 million USD and $174 million USD (2016 data) [135]. On the organizational level, more spaced-out courses of immunotherapy, e.g., three- or four-week intervals instead of every two weeks, may improve organization in hospital pharmacies and frequency of patient visits, with subsequent reduced transportation cost. Modification of immunotherapy scheduling could ultimately lower costs for cancer institutes, insurance companies and social security systems. In Europe during the COVID-19 period, clinicians and institutions have been seeking opportunities to limit exposure to SARS-CoV-2 by reducing patient visits to hospital. National recommendations in France were to increase dosing (4 mg/kg) and to space out courses of treatment (four or six-weekly administration) for anti-PD-1 in monotherapy [136]. Goldstein concluded that weight-based dosing of pembrolizumab every six weeks made rapid re-organization possible, with equivalent efficacy, decreased financial impact and risk of SARS-Cov-2 exposure [137]. It would therefore seem essential to take into account the previously presented data to define individualized care for these increasingly used treatments. For example, in France alone, 3966 new patients received nivolumab in 2015, and 8720 received nivolumab in 2016; 903 and 1023 new patients received pembrolizumab in 2015 and 2016, respectively, thus highlighting the constant increase in patients treated with ICIs [138].

## 9. Conclusions

In less than a decade, immunotherapy has become a new cornerstone in cancer treatment, along with chemotherapy, surgery, radiotherapy, and targeted therapy. Indeed, it has improved clinical outcomes, with profound and sustainable response for a subset of patients with once dismal prognosis. However, it has also changed many paradigms in drug development, from the necessity for specific murine in vivo models in preclinical studies to the poor PK/PD relationships enabling the use of flat doses for PD-1/PD-L1 inhibitors. The development of ICIs has relied strongly on the application of quantitative methods and in silico simulations for decision making. As such, it is as a remarkable example of model-informed drug development. This development has increased our knowledge of mAb PK, and many principles in term of interpreting PK/PD relationships may even be generalized to other mAbs, especially those used in targeted cancer therapy. Unlike cytotoxics or oral kinase inhibitors, using TDM in order to improve efficacy or tolerance seems unlikely. But monitoring ICI concentrations could be implemented and evaluated with the perspective of postponing infusion if sufficient concentrations are measured in a patient, with marked impact in terms of drug-cost. Indeed, ICIs can probably be administered at a lower dose and/or according to a longer inter-cycle period without compromising their efficacy. However, these changes must be supported by further prospective PK and PK/PD evaluations in real-world settings.

## Figures and Tables

**Table 1 vaccines-08-00632-t001:** Usual doses and clinical applications of Food and Drug Administration (FDA) approved immune check-point inhibitors (ICIs). CSCC, cutaneous squamous cell carcinoma; NSCLC: non-small cell lung cancer; TNBC, triple-negative breast cancer.

Target	Drug	Body-Weight-Based Dose	Flat Dose	Clinical Applications
CTLA-4	Ipilimumab (YERVOY^®^)	3 mg/kg Q3W 10 mg/kg Q3W		Metastatic melanoma Cutaneous melanoma Advanced renal cell carcinoma.
PD-1	Nivolumab (OPDIVO^®^)	3 mg/kg Q2W	240 mg Q2W 480 mg Q4W	Metastatic melanoma Metastatic NSCLC Hodgkin lymphoma Advanced renal cell carcinoma Advanced or metastatic urothelial carcinoma Metastatic colorectal cancer Hepatocellular carcinoma
	Pembrolizumab (KEYTRUDA^®^)	2 mg/kg Q3W	200 mg Q3W 400 mg Q6W	Melanoma NSCLC Head and neck squamous cell cancer Classical Hodgkin lymphoma Primary mediastinal large b-cell lymphoma Urothelial carcinoma Microsatellite instability-high cancer Gastric cancer Cervical cancer Hepatocellular carcinoma Merkel cell carcinoma
	Cemiplimab (LIBTAYO^®^)		350 mg Q3W	Metastatic CSCC Locally advanced CSCC
PD-L1	Atezolizumab (TECENTRIQ^®^)		840 mg Q2W 1200 mg Q3W 1680 mg Q4W	Urothelial Carcinoma NSCLC TNBC Metastatic treatment of TNBC
	Avelumab (BAVENCIO^®^)	10 mg/kg Q2W	800 mg Q2W	Metastatic Merkel cell carcinoma Advanced or metastatic urothelial carcinoma Advanced renal cell carcinoma (+axitinib)
	Durvalumab (INFINZI^®^)	10 mg/kg Q2W	750 mg Q2W 1500 mg Q4W	Locally advanced or metastatic urothelial carcinoma Unresectable stage III NSCLC

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
