# Peer review of "Cancer Immunotherapy Dosing: A Pharmacokinetic/Pharmacodynamic Perspective"

_vaccines, 2020, doi:10.3390/vaccines8040632_

Round 1
Reviewer 1 Report
In this review article the authors describe pharmacokinetc and pharmacodynamic aspects concerning the required and adequate doses of different immunotherapeutic products (and their combination) used as check-point inhibitors (CPI) effective upon different types of cancer diseases. It seems that the subject concerning this study may present specific challenges since differently from common safety procedures applied to usual drug treatment biological products are well tolerated. Thus, as an important conclusion, the authors stressed that "CPIs can probably be administrated at a lower dose and/or according to a longer inter-cycle period without compromising their efficacy". The work is well written and present interesting approaches to face this challenging and tricking problem and thus it deserves publication. Nevertheless some small points in the manuscript must be clarified.
- Since a great number of abbreviations were used, an abbreviation session should be included in the manuscript;
- Lines 537-538: the sentence "Radiation has also been also associated..." must be rewritten;
- Lines 679 -680: the sentence "...and is serves as a remarkable..." also requires correction.
Author Response
Reviewer’s comments in italic
Answers to Reviewer 1
In this review article the authors describe pharmacokinetc and pharmacodynamic aspects concerning the required and adequate doses of different immunotherapeutic products (and their combination) used as check-point inhibitors (CPI) effective upon different types of cancer diseases. It seems that the subject concerning this study may present specific challenges since differently from common safety procedures applied to usual drug treatment biological products are well tolerated. Thus, as an important conclusion, the authors stressed that "CPIs can probably be administrated at a lower dose and/or according to a longer inter-cycle period without compromising their efficacy". The work is well written and present interesting approaches to face this challenging and tricking problem and thus it deserves publication. Nevertheless some small points in the manuscript must be clarified.
We do thank you for your interest in this work and your valuable comments.
- Since a great number of abbreviations were used, an abbreviation session should be included in the manuscript;
Following your recommendations, an abbreviation section has now been added (Pages 2-4) and abbreviations were further checked and homogenized throughout the manuscript.
- Lines 537-538: the sentence "Radiation has also been also associated..." must be rewritten;
The sentence was re-written and a reference was added.
- Lines 679 -680: the sentence "...and is serves as a remarkable..." also requires correction.
The typos have been corrected.
Reviewer 2 Report
In this manuscript, the authors indicated that the check-point inhibitors (CPI) are widely used in the clinic. Different doses of CPIs have different pharmacodynamic variability. To better understand whether or not dosing could be an actionable item in oncology, preclinical and clinical development of CPI were described. First, the authors described the challenges of preclinical models in the era of immunotherapy. There are marked differences between murine mAbs and human mAbs in terms of pharmacokinetics and target engagement. To resolve this inherent difficulty, sophisticated animal models, such as human Knock-In (KI) mice, immuno-avatar mice, or hemato-lymphoid humanized mice, have been developed. However, each of the above models has its own flaws. Then, the authors researched early clinical trials and dose findings and analyzed the therapeutic effect of different doses of CPIs. Subsequently, the authors demonstrated the adverse events of applying CPIs, and the so-called immune-related adverse events (irAEs) are now well described clinically, but the predominant mechanism of action and the long-term effects on patients' immunity remain unclear. Furthermore, the authors pointed out the combinations represent a key issue and indicated the applications of different combinations with CPIs, cytotoxics, targeted therapy, radiotherapy, and other combinations in various cancers. Finally, the authors concluded that the development of CPIs has relied strongly on the application of quantitative methods and in silico simulations for decision-making and serves as a remarkable example of model-informed drug development. However, a few issues still need to be clarified prior to the further consideration of this manuscript.
- The authors mentioned in the introduction to review the preclinical and clinical development of the consumer price index from the perspective of dose research. However, in the second part, there is no mention of issues related to dose. So, the reviewer suggests that the authors should analyze the problem more from the perspective of dose.
- The authors take immune check-point inhibitors as the research object to analyze pharmacokinetics and pharmacodynamics. However, the authors only described PD-1 and CTLA-4. The review suggests that the authors could add some other applications of CPIs to better illustrate your viewpoint, such as phagocytosis check-points.
- In the "Combinations represent a key issue" part, the authors listed different combinations to treat cancer. However, the authors only explained the advantages of different combinations, but did not use data to illustrate their pharmacokinetic characteristics. So, the reviews suggest that the authors should describe the pharmacokinetic in more detail for readers to understand the article more comprehensively.
- In the "Flat dose and modified schedules" part, the authors indicated that side effects were poorly related to the dose. This statement is contrary to people's perception. So, the reviews suggest that the authors could check whether the statement is correct and describe what the side effects are related to.
- The references should be updated. The recently published review and research articles should be discussed in the revision, for example, Advanced Science 2019, 6 (17), 1900101.
Author Response
Reviewer’s comments in italic
Answers to Reviewer 2
In this manuscript, the authors indicated that the check-point inhibitors (CPI) are widely used in the clinic. Different doses of CPIs have different pharmacodynamic variability. To better understand whether or not dosing could be an actionable item in oncology, preclinical and clinical development of CPI were described. First, the authors described the challenges of preclinical models in the era of immunotherapy. There are marked differences between murine mAbs and human mAbs in terms of pharmacokinetics and target engagement. To resolve this inherent difficulty, sophisticated animal models, such as human Knock-In (KI) mice, immuno-avatar mice, or hemato-lymphoid humanized mice, have been developed. However, each of the above models has its own flaws. Then, the authors researched early clinical trials and dose findings and analyzed the therapeutic effect of different doses of CPIs. Subsequently, the authors demonstrated the adverse events of applying CPIs, and the so-called immune-related adverse events (irAEs) are now well described clinically, but the predominant mechanism of action and the long-term effects on patients' immunity remain unclear. Furthermore, the authors pointed out the combinations represent a key issue and indicated the applications of different combinations with CPIs, cytotoxics, targeted therapy, radiotherapy, and other combinations in various cancers. Finally, the authors concluded that the development of CPIs has relied strongly on the application of quantitative methods and in silico simulations for decision-making and serves as a remarkable example of model-informed drug development. However, a few issues still need to be clarified prior to the further consideration of this manuscript.
We do thank you for your interest in this paper and your valuable comments.
- The authors mentioned in the introduction to review the preclinical and clinical development of the consumer price index from the perspective of dose research. However, in the second part, there is no mention of issues related to dose. So, the reviewer suggests that the authors should analyze the problem more from the perspective of dose.
I am afraid this is some kind of misunderstanding. CPI means Check Point Inhibitors and definitively not Consumer Price Index. We have now replaced the “CPI” acronym by the more widely used “ICI” (Immune Checkpoint Inhibitors) acronym.
- The authors take immune check-point inhibitors as the research object to analyze pharmacokinetics and pharmacodynamics. However, the authors only described PD-1 and CTLA-4. The review suggests that the authors could add some other applications of CPIs to better illustrate your viewpoint, such as phagocytosis check-points.
A paragraph was added in the appropriate section to describe checkpoint inhibitors targeting other axes of tumor immunity as requested (Page 11, lines 274-286). Still, please note that as of today, the only approved drugs interfere only with CTLA4 or the PD1/PDL1 axis. In this report, we primarily focused on currently approved drugs with a body of literature sufficient to support our claims. Because new drugs (such as anti-IDO) have not reached yet late clinical studies phases, little is now about PD endpoints.
- In the "Combinations represent a key issue" part, the authors listed different combinations to treat cancer. However, the authors only explained the advantages of different combinations, but did not use data to illustrate their pharmacokinetic characteristics. So, the reviews suggest that the authors should describe the pharmacokinetic in more detail for readers to understand the article more comprehensively.
A new paragraph describing the pharmacokinetic characteristics of mAbs has now been added following your recommendations (Page 21, lines 534-538). Please note most combinatorial strategies with immunotherapy totally skip the PK issue – therefore, little is known regarding the exact PK/PD relationships of the combined treatment.
- In the "Flat dose and modified schedules" part, the authors indicated that side effects were poorly related to the dose. This statement is contrary to people's perception. So, the reviews suggest that the authors could check whether the statement is correct and describe what the side effects are related to.
Indeed, checkpoint inhibitors’ adverse events are poorly related to their dose, mostly because treatment-related toxicities are primarily dependent on each patient’s innate immunity and susceptibility, instead of being driven by the Hill equation as most other drugs in oncology. For instance, no Maximum Tolerated Dose has been found during dose-ranging studies with all the approved drugs, as underlined in the manuscript. This point has been better detailed (Page 13, lines 338-342) and new references were added as requested. Of note, full details regarding the underlying mechanisms of toxicities are provided.
- The references should be updated. The recently published review and research articles should be discussed in the revision, for example, Advanced Science 2019, 6 (17), 1900101.
The reference section has been updated (new references: 1-3,29-33, 86,87,89,103-105).
Reviewer 3 Report
This is a comprehensive and well complied review article. Minor suggestions for improvements are as follows:
- The authors should also discuss limitations associated with the use of CPIs such as toxicity, drug resistance etc.
- A separate table should be provided to list the names, mechanisms of actions and clinical applications of various FDA approved CPIs.
- The authors should provide their own justification and relevance of the study. This will help the readers to understand the importance of the paper.
- Typographical errors were found throughout the manuscript and should be corrected.
Author Response
Reviewer’s comments in italic
Answers to Reviewer 3
This is a comprehensive and well complied review article. Minor suggestions for improvements are as follows:
We do thank you for your interest in this MS and your valuable comments.
- The authors should also discuss limitations associated with the use of CPIs such as toxicity, drug resistance etc.
Toxicity and its relationship to drug dosing are now been completed (Page 13, lines 338-342). Comments relative to drug resistance of CPI were also added at the beginning of Section 7 (Page 21, lines 523-528). However, unlike with other anticancer drugs such as cytotoxics, toxicity is not a major clinical issue with single agent immunotherapy and is not correlated with overexposure and therefore dosing issues. Similarly, drug resistance with immune checkpoint inhibitors is most probably due to “immune desert” condition and immunologically cold tumors, making ICIs ineffective. This is why most biomarkers proposed thus far such as PDL1 expression, Tumor mutational Burden, or neutrophils-to-lymphocytes ratios are related to harnessing tumor immunity, and not on exposure-effect or dosing issues. In this respect, and because this paper mostly focuses on the issue of dose-finding with immunotherapy, concerns with treatment-related toxicities are not the primary issue here.
- A separate table should be provided to list the names, mechanisms of actions and clinical applications of various FDA approved CPIs.
Thanks for this valuable suggestion. A Table describing targets, usual doses and clinical applications of various FDA approved CPIs is now added (Pages 14-15) in Section 4 following your recommendations.
- The authors should provide their own justification and relevance of the study. This will help the readers to understand the importance of the paper.
This is now better underlined in the revised version of the MS, i.e. in Introduction and Conclusion sections, especially regarding the context of dose individualization and therapeutic drug monitoring in oncology.
- Typographical errors were found throughout the manuscript and should be corrected
Manuscript has been corrected for typos.